# A 30-m annual maize phenology dataset from 1985 to 2020 in China

Quandi Niu[1], Xuecao Li[1,2], Jianxi Huang[1,2*], Hai Huang[1], Xianda Huang[1], Wei Su[1,2], Wenping Yuan[3]

[1] College of Land Science and Technology, China Agricultural University, Beijing 100083, China

[2] Key Laboratory of Remote Sensing for Agri-Hazards, Ministry of Agriculture and Rural Affairs, Beijing 100083, China

[3] School of Atmospheric Sciences, Sun Yat-sen University, Guangzhou 5120245, Guangdong, China

*Correspondence to*: Jianxi Huang (jxhuang@cau.edu.cn)

**Abstract.** Crop phenology information provides essential information on crop growth phases, which are highly required for agroecosystem management and yield estimation. Previous crop phenology studies were mainly conducted using coarse-resolution (e.g., 500 m) satellite data, such as the moderate resolution imaging spectroradiometer (MODIS) data. However, precision agriculture requires higher resolution phenology information of crops for better agroecosystem management, and this requirement can be met by long-term and fine-resolution Landsat observations. In this study, we generated the first national maize phenology product with a fine spatial resolution (30m) and a long temporal span (1985-2020) in China, using all available Landsat images on the Google Earth Engine (GEE) platform. First, we extracted long-term mean phenological indicators using the harmonic model, including the v3 (i.e., the date when the third leaf is fully expanded) and the maturity phases (i.e., when the dry weight of maize grains first reaches the maximum). Second, we identified the annual dynamics of phenological indicators by measuring the difference of dates when the vegetation index in a specific year reaches the same magnitude as its long-term mean. The derived maize phenology datasets agree with in-situ observations from the agricultural meteorological stations and the PhenoCam network. Besides, the derived fine-resolution phenology dataset agrees well with the MODIS phenology product regarding their spatial patterns and temporal dynamics. We observed a noticeable difference in maize phenology temporal trends before and after 2000, which is likely attributable to the change of temperature and precipitation, which further alter the farming activities. The extracted maize phenology dataset can support precise yield

estimation and deepen our understanding of the response of agroecosystem to global warming in the future. The data are available at https://doi.org/10.6084/m9.figshare.16437054 (Niu et al., 2021).

## 1 Introduction

Accurate and timely crop phenology information, which contains multi-phase growth information from sowing to harvest, is highly required by precision agriculture management (Gao and Zhang, 2021; Zeng et al., 2020), such as irrigation schedules and pest control. The agriculture management schemes should be precisely scheduled according to different growth phases, during which period the water requirements and the possibilities of pest and disease events are different (Yang et al., 2021; Xiao et al., 2020). Besides, the effect of climate change on crop phenology has been widely reported (Abbas et al., 2017; Zhang and Tao, 2013; Tao et al., 2012), given that the altered growth phases of crops will influence crop production. Thus, further research into the response of crop phenology to global warming is necessary, which requires long-term records of phenology change. In addition, information on crop phenology is also helpful for crop mapping because different crops vary in their growth phases (Sakamoto et al., 2014; Zhong et al., 2014; Zhang et al., 2014; Huang et al., 2019b).

Remote sensing has become a profound tool to derive crop phenology at a large scale (Pan et al., 2015; Liu et al., 2018). The annual variations of crop phenology are affected by many factors, including climate conditions, soil properties, and anthropogenic activities (e.g., sowing dates) (He et al., 2020). The traditional in-situ based crop phenology recording is time-consuming and focused on limited sites (Gao and Zhang, 2021). These limitations have been considerably mitigated by satellite images, which provide revisit observations of crop growth at regional and global scales (Shanmugapriya et al., 2019; Zhang et al., 2003; Cao et al., 2015). Different phenological indicators (such as the start of season and the end of season) are retrieved for crop growth monitoring using satellite observations, including the moderate resolution imaging spectroradiometer (MODIS) data (Sakamoto et al., 2010), the advanced very high resolution radiometer (AVHRR) data (Zhang et al., 2014; Gim et al., 2020). The retrieved multiple phenological indicators can delineate the development stages of crops from sowing to harvest at a regional and global scale.

Fine resolution Landsat satellite data show great potential in providing crop phenological indicators with a fine resolution and a long-term span. Despite that those coarse satellite data (such as MODIS and AVHRR) have a fine temporal resolution, which

is helpful to depict the crop growth phases, they are limited in their spatial resolution. Recently, several attempts have been made at deriving phenology datasets using fine resolution satellite data, such as Landsat (Li et al., 2019; Senf et al., 2017), Sentinel-2 (Bolton et al., 2020), and the harmonized Landsat8 and Sentinel-2 (HLS) data (Claverie et al., 2018; Bolton et al., 2020). Compared with medium-resolution satellite data such as MODIS, the Landsat satellite data can provide abundant land

surface records from 1985 to the present, which help derive the long-term crop phenology dynamics. Unfortunately, limited attempts have been made using Landsat data to map the crop phenology with a fine resolution and a long-term span in China due to the complex planting patterns (Luo et al., 2020; Wu et al., 2010). Also, the computing resources required for such a mapping project are a huge challenge(Dong et al., 2016).

The advent of the Google Earth Engine (GEE) platform relieves the huge stress of data storage and computing at regional and

global scales. The GEE platform has included petabyte-scale remote sensing data with high-performance computing capabilities and powerful algorithm libraries (Gorelick et al., 2017). Presently, many successful attempts have been conducted using the GEE platform, such as mapping of forest dynamics (Xiong et al., 2020), terrace (Cao et al., 2021), and surface water (Pekel et al., 2016). It is convenient to obtain and process satellite data using the GEE platform. The combination of massive satellite observations and a flexible development environment makes it possible to derive annual dynamics of crop phenology

with fine resolution in China.

In this study, we extracted spatial and temporal patterns of maize phenology indicators in China from Landsat observations using the GEE platform. The derived phenology indicators include v3 (the date when the third leaf is fully expanded) and maturity (i.e., when the dry weight of maize grains first reaches the maximum) phase. We mapped annual phenological indicators of maize at a fine resolution (30m) from 1985 to 2020, using full achieve of Landsat images. The remainder of this

paper is organized as below: Section 2 introduces the study area and datasets, Section 3 presents the method used in this study, Section 4 and Section 5 describes the results with discussion and the derived dataset, respectively, and an ending mark is provided in Section 6.

## 2 Study area and datasets

 In this study, we selected China's main maize producing area as our study area (Fig. 1). Maize is one of the major crops in China and planted over a wide region, the sown area and production account for 36% and 39% of food crops in 2019 (China Statistical Yearbook, 2020), respectively. The planting pattern and phenology of maize are highly heterogeneous due to the

5 influence of climate conditions, soil properties, and anthropogenic activities (e.g., sowing date)(Wu et al., 2010). The spring maize is mainly distributed in Northeast China, dominated by single cropping due to the accumulated temperature limit. However, summer maize is mainly planted in the Huang-Huai-Hai Plain, dominated by the double cropping system of winter wheat-summer maize rotation(Luo et al., 2020). Besides, other provinces (e.g., Xinjiang province) also have a certain distribution of maize. Under these diverse cropping systems, phenology dates (such as v3 and maturity) of maize varied

significantly over space.

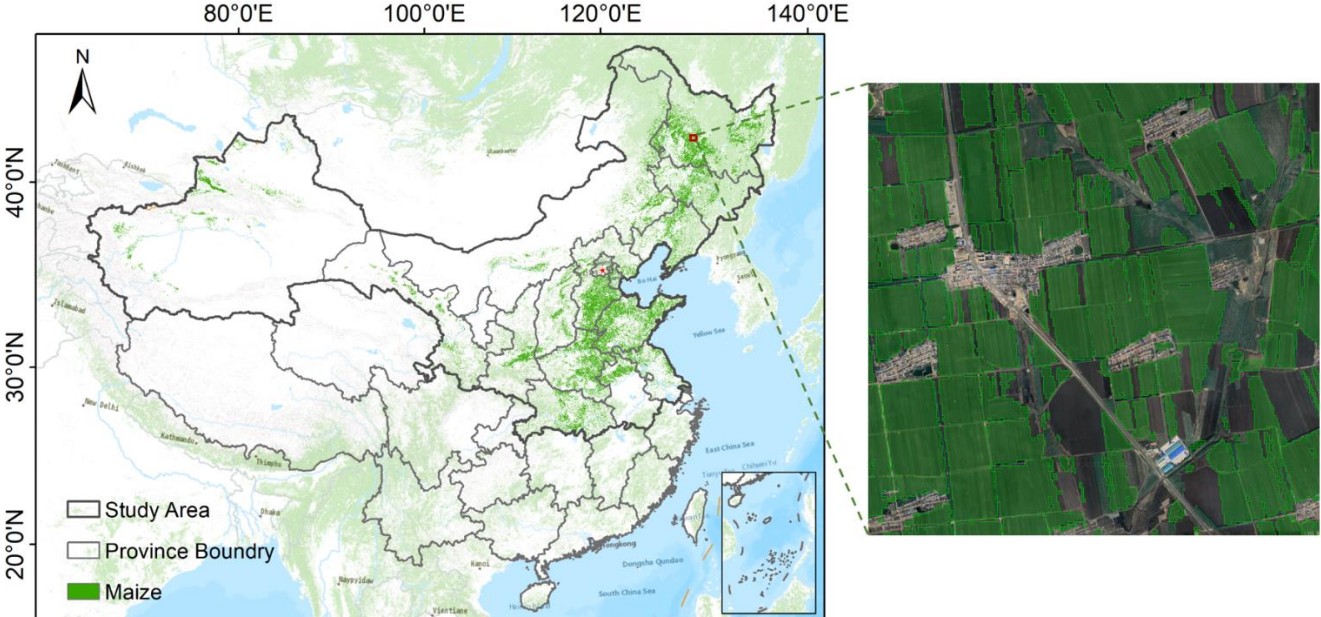

**Fig. 1:** Distribution of maize within study area, the basemap is from ESRI (https://www.arcgis.com/apps/mapviewer/index.html).

We used the Landsat satellite data as the primary data source to characterize the phenology changes of maize in China. We used all available Landsat surface reflectance data in this study, including images obtained from Thematic Mapper (TM),

Enhanced Thematic Mapper Plus (ETM+), and Operational Land Imager (OLI), from 1985 to 2020. The Landsat surface reflectance data have been corrected for the radiometric and topographic effects. The atmospheric effect has been corrected

using the Landsat ecosystem disturbance adaptive processing system (LEDAPS) (Masek et al., 2006). Clouds and shadows were removed using the function of the mask procedure (Zhu and Woodcock, 2012). Therefore, all available clear-sky pixels of Landsat observations over past decades were used in our study.

Maize maps from multiple resources were adopted to constrain the region of crop phenology mapping. The distribution map

of maize in Northeast China was derived using Sentinel-2 data (You et al., 2021), and the used maize map was the classification result in 2019. While in other provinces, the maize maps were obtained using the method in Dong et al. (2020) to process Landsat images in 2020. The accuracy of the maize map in Northeast China is 0.85, and that of maize maps in other provinces is 0.79. Given that the original resolutions of these two classification maps are 10m (i.e., Northeast China) and 30m (i.e., other provinces), we resampled them to the same resolution of 30 m. It is worth note the maize distribution map is consistent across

different years in our study due to the relatively stable planting situation(Sun et al., 2007; Li et al., 2008).

In addition, we also collected other datasets to validate our results, such as the agricultural meteorological stations (AMS), PhenoCam network, and the MODIS phenology product (MCD12Q2). First, the records in AMS include phenology information of major crops (such as maize, wheat, and rice) in China, with large spatial and temporal ranges (Luo et al., 2020; Huang et al., 2019a). Crucial phases during the maize growth, including v3 (i.e., the date when the third leaf is fully expanded),

and maturity (when the dry weight of maize grains first reaches the maximum) phases, were recorded in the AMS. Thus, this dataset can validate the mapped phenological indicators from remote sensing, and we collected AMS phenology records of the spring and summer maize (Fig.2). Second, the in-situ PhenoCam observation was derived from digital cameras using the green chromatic coordinate (GCC) indicator, which is composited by visible wavebands and able to characterize the dynamic greenness of vegetation. Third, the MODIS phenology product (MCD12Q2) was also employed in our study to validate the

results derived from Landsat observations. The phenological indicators (e.g., dates of greenup and dormancy) in the MODIS product were mainly derived from the two-band enhanced vegetation index (EVI2) time series data (Gray et al., 2019). The multiple cycles (up to two) of crop rotations were also recorded in the MODIS phenology product, which is suitable for validation with our phenology results of maize in this study.

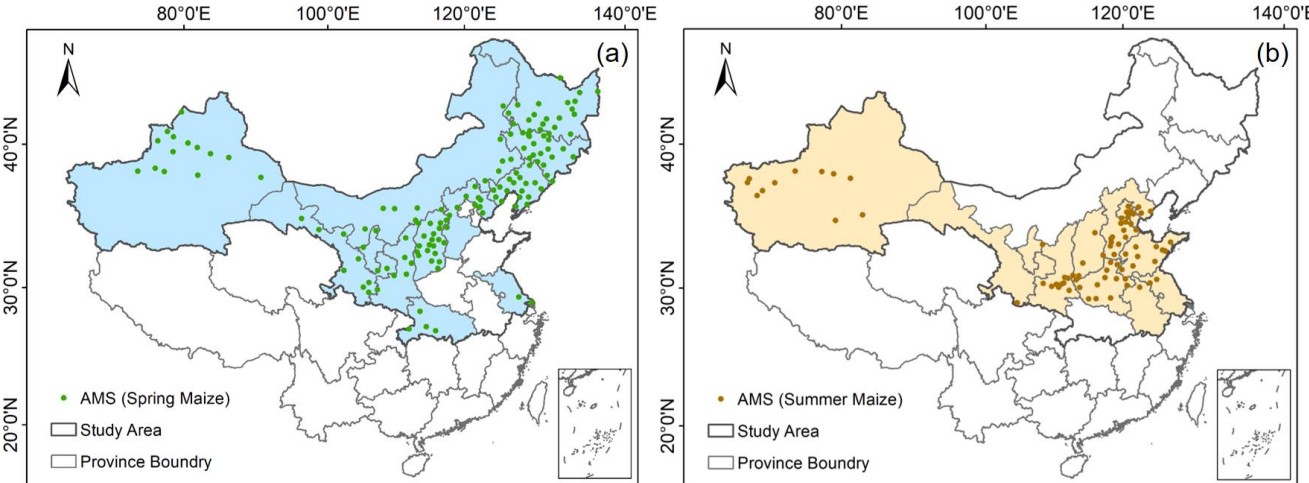

**Fig. 2:** Distribution of agricultural meteorological stations (AMS) with phenology records of spring (a) and summer (b) maize. The blue and yellow areas are provinces with available AMS in this study.

## 3 Methodology

5   We extracted the phenology indicators of maize using the full archive of Landsat images in GEE. The adopted framework includes three components (Fig. 3). First, we collected all available Landsat pixels during 1985-2020 in our study area using the collected maize map as a mask. After the cloud removal, we constructed the long-term time series data of EVI for each pixel. Second, we fitted the long-term mean EVI curve using the harmonic model, which identifies spring and summer maize. Thus, two phenological indicators, the v3 (the date when the third leaf is fully expanded) and the maturity (when the dry weight of maize grains first reaches the maximum) phase, were determined from the long-term mean curve of spring and summer
10  maize. Finally, we identified the annual dynamics of these two phenology indicators by measuring the difference of dates when the vegetation index in a specific year reaches the same magnitude as its long-term mean. Details of each component were given in the following sections.

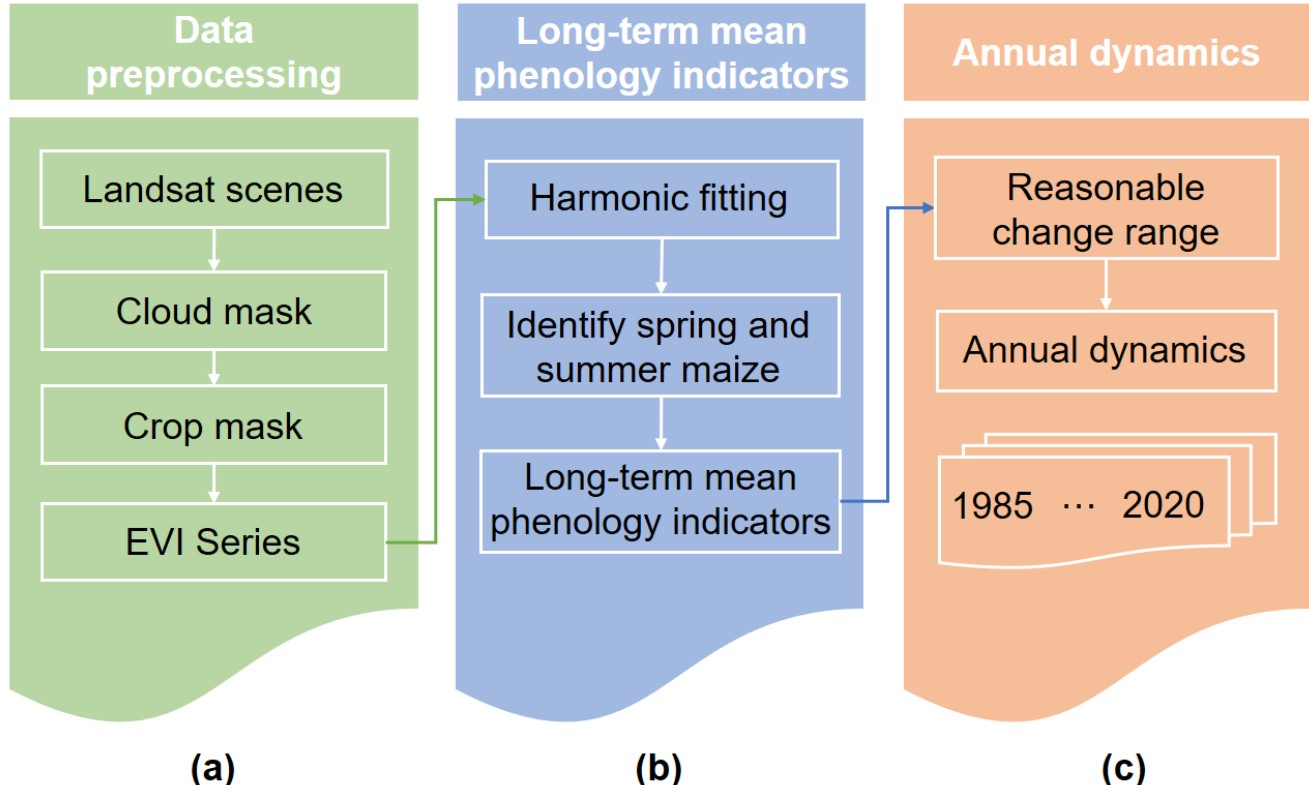

**Fig. 3:** The adopted framework for deriving annual phenology dynamics (1985-2020) from Landsat time series data, including data preprocessing (a), mapping the long-term mean phenology indicators (b), and identifying the annual dynamics (c).

### 3.1 Data preprocessing

5   We implemented the data preprocessing step in the GEE platform. First, we used the quality layer in the Landsat surface reflectance data to remove clouds and shadows. Thus, all available clear-sky pixels can be used to enrich the Landsat observations. Second, we excluded non-maize areas using the maize map, which can significantly reduce the computational requirement. Third, we calculated the EVI indicator using Eq. (1) to minimize the impact of soil and clouds; meanwhile, it is sensitive to vegetation growth and dormancy (Huang et al., 2019a; Li et al., 2019).

$$EVI = G * \frac{NIR - RED}{NIR + C_1 * RED - C_2 * BLUE + L} \qquad (1)$$





where NIR, RED, BLUE represent surface reflectance of the corresponding spectral bands in Landsat. Parameters of G, L, $C_1$, $C_2$ were used to correct the disturbance of aerosols and soil background, as suggested with values of 2.5 (G), 1(L), 6($C_1$), 7.5($C_2$) in Huang et al. (2019a).

**3.2 Long-term mean phenology indicators**

5   Using the harmonic model, we derived the long-term mean maize phenology indicators (e.g., v3 and maturity). First, we sorted all available EVI observations according to their day of the year (DOY) and fitted the annual crop cycle using the harmonic model (Eq. 2). Comparing to other fitting approaches, the harmonic model can well delineate multiple seasonal cycles of crops within one year, with clear physical meanings of each parameter (de Beurs and Henebry, 2010; Chen et al., 2018; Lee et al., 2020).

$$f(t) = a_0 + a_1 \frac{t}{T} + \sum_{i=1}^{n} \left( b_i \cos\left(i \frac{2\pi t}{T}\right) + c_i \sin\left(i \frac{2\pi t}{T}\right) \right) \tag{2}$$

where f(t) is the fitted EVI value, t is the Julian date of a particular observation, and T is the maximum value of the time variable. bi and ci are coefficients for intra-annual change of the EVI time series data. a1 and a0 represent the slope and intercept of EVI change among different seasonal cycles. n represents the maximum number of harmonic components, and it needs to be calibrated according to different situations. Considering the double-crop (winter wheat-summer maize rotation

15   system) and the planting patterns of winter wheat (planted in autumn of the first year and harvested in the second year), we set n as 6 in our study after trial and error test using multiple sites in different regions, due to the good fitting performance.

Then, we identified spring and summer maize according to the cycles of the fitted curve (Fig. 4). Spring and summer maize can be identified using the information of EVI cycles. For instance, summer maize always has two crop cycles, notably different from spring maize with only one cycle. To identify maize with different cycles, we calculated the derivative of the

20   fitted harmonic model and identified the peaks of these cycles. When the EVI peak value before the maize part exceeded 40% of the maximum EVI value of the maize cycle (Gray et al., 2019; Wu et al., 2010), we regarded it as double cropping, and the second part of it is summer maize (Fig. 4b); otherwise, it is a single crop (i.e., spring maize) (Fig. 4a).



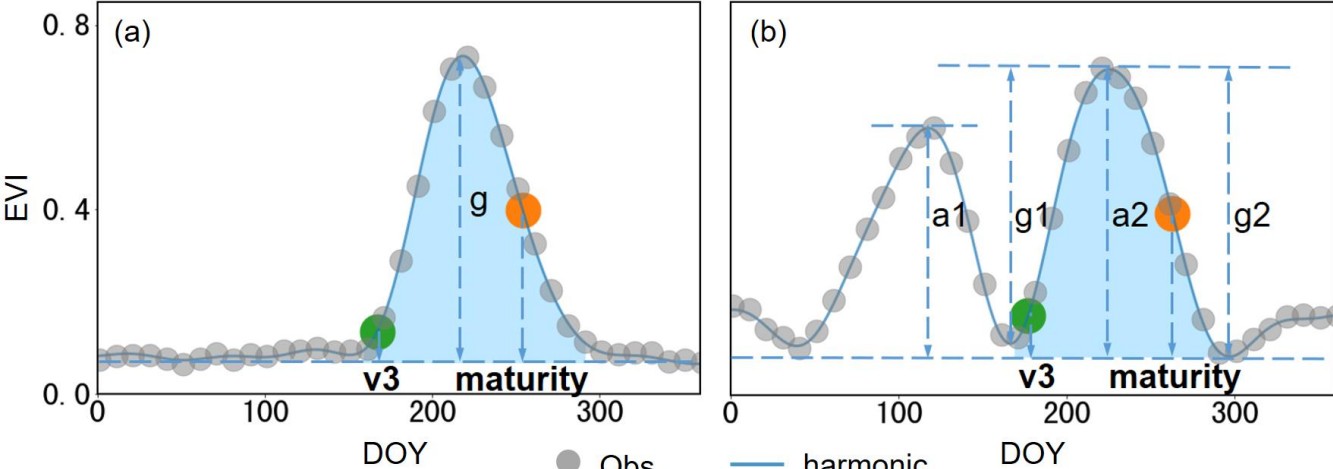

**Fig. 4:** Illustration of mapping long-term mean phenology of spring maize (a) and summer maize (b). The blue shaded areas represent the growing period of maize. The definition of all acronyms are as follows: g: the EVI amplitude of spring maize; g1 and g2 are the EVI amplitudes of green-up and green-down segments, respectively; a1 and a2 are the EVI amplitudes of the first and second cycle; DOY: day of year.

Finally, we adopted the dynamic threshold approach to derive the v3 and maturity dates from the green-up and green-down segments (shadowed blue areas in Fig. 4), respectively. These two segments were derived from the cycle of maize growth, separating by the point with peak values of EVI. Given that the EVI amplitudes of green-up and green-down are different for the spring (g in Fig. 4a) and summer maize (g1 and g2 in Fig. 4b), we determined the v3 and maturity dates according to their EVI amplitudes accordingly. For the spring maize, the v3 and maturity dates were defined as the dates with 10% EVI amplitude (g in Fig. 4a) during the green-up segment and 50% EVI amplitude (g in Fig. 4a) during the green-down segment (Huang et al., 2019a), respectively. Similarly, for the summer maize, the EVI amplitude during the green-up and green-down segments were referred to g1 and g2 in Fig. 4b, for the determination of v3 and maturity, respectively.

### 3.3 Annual dynamics of phenology indicators

We adopted a similar approach in Li et al., (2017) to estimate the annual dynamics of phenology indicators. First, we adopted a self-adjusting strategy to determine the rational range of EVIs during the green-up and green-down periods (shaded areas in Fig. 5). These ranges were determined using the derived long-term mean curve, and they can be used to filter outliers in



individual years. Then, we measured the difference of dates when the EVI in a specific year reaches the same magnitude as its

long-term mean (Fig. 5). The mean value of date difference between the observations and the long-term mean was adopted as

the annual variability of phenological indicators.

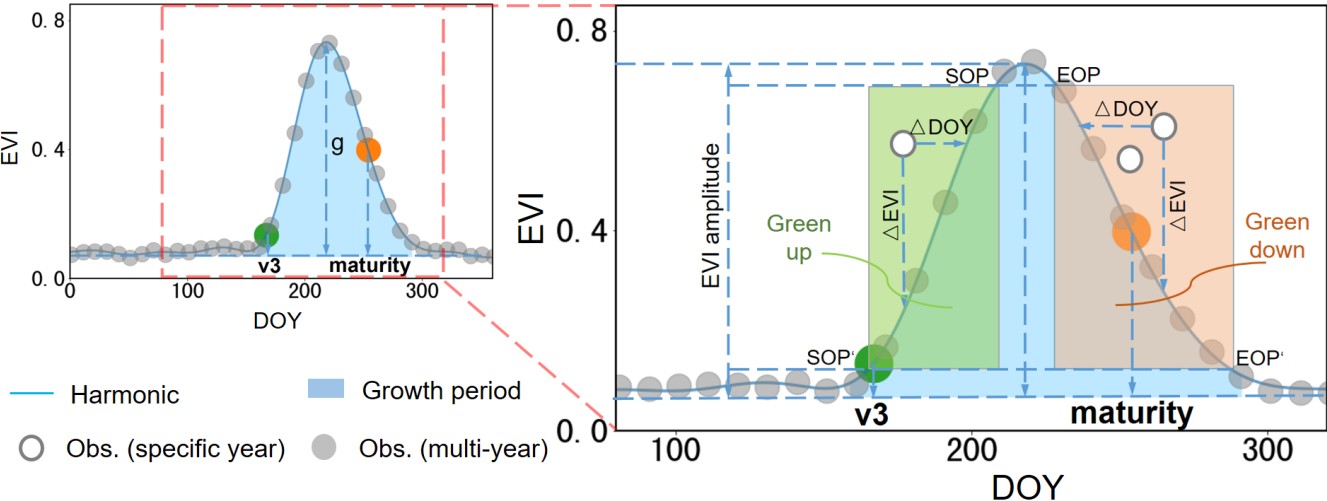

**Fig. 5:** Illustration of detecting annual variabilities of phenological indicators (taking the spring maize as an example). The blue shaded areas

represent the growth period of maize. In addition, green and orange shaded areas indicate the reasonable change range of v3 and maturity.

The solid and empty circles are long-term and year-specific enhanced vegetation index (EVI) observations, respectively. The definition of

all the acronyms are as follows: EOP: end of peak; SOP: start of peak.

## 4 Results and discussion

### 4.1 Performance of the harmonic model

The harmonic model can well delineate the seasonal dynamics of EVI for spring and summer maize. Spring maize always has

one crop cycle, whereas there are two cycles in the summer maize, of which the second one is mainly caused by crop rotation

(i.e., winter wheat and summer maize). These crop growth cycles can be well detected by the EVI time series data from Landsat

observations (Fig. 6). The fitting performance from the harmonic model suggests the fitted line can well delineate the growth

phase of crops across different regions and types.

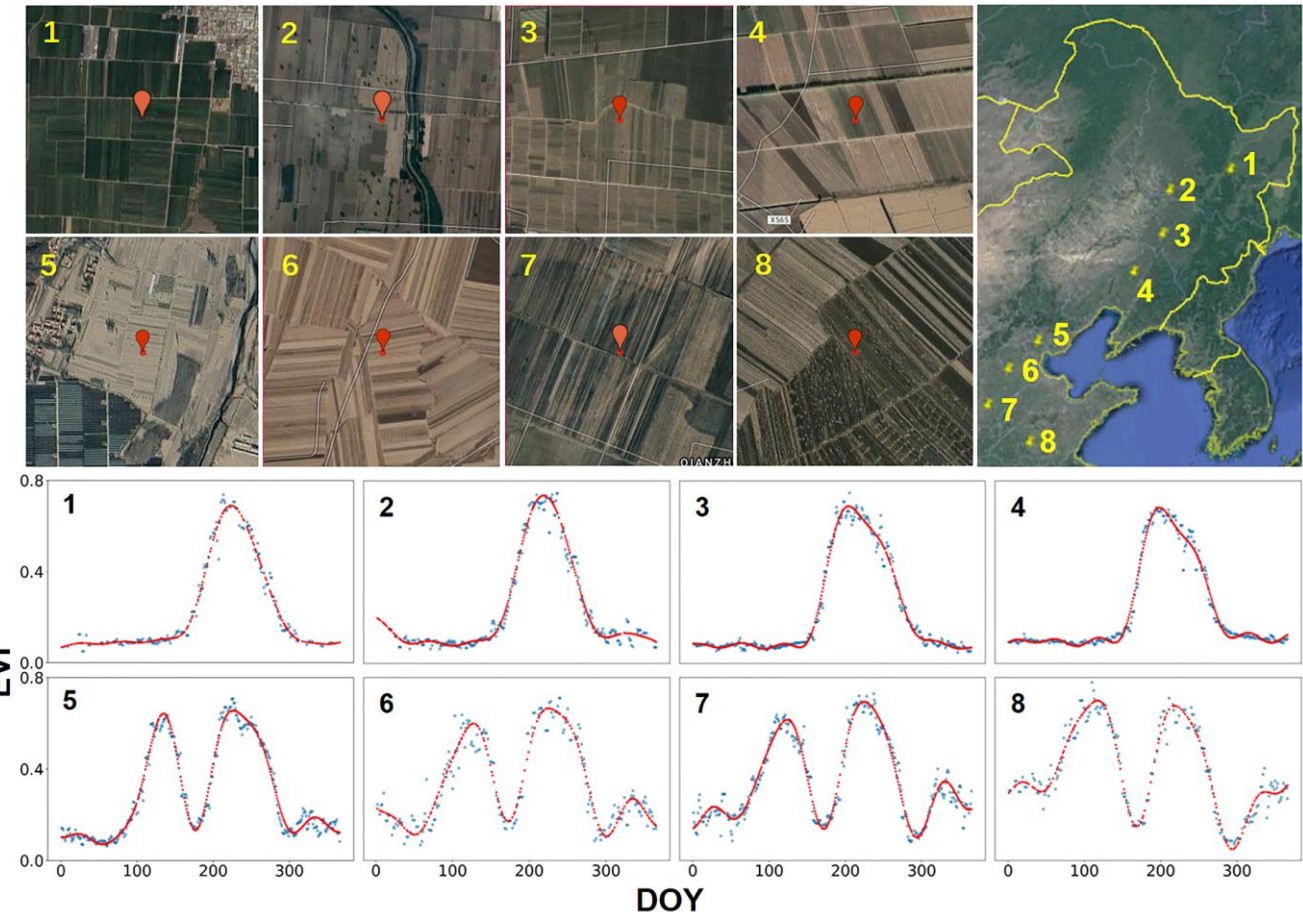

**Fig. 6:** Performance of harmonic model in fitting the time series data of EVI. Cases numbered from 1-4 and from 5-8 represent the spring and summer maize, respectively. The blue points are the original EVI series of red dots in the center of ⓒ Google Earth images and the red points are the fitted series.

## 4.2 Comparison with records from the AMS

The derived long-term mean maize phenology indicators from Landsat observations are consistent with the records from the AMS (Fig. 7). We compared results derived from Landsat and AMS from 2001 to 2010, of which period the AMS observations can be maximally used. Due to the lack of accurate locations of observed crops in AMS (i.e., only station locations), we measured the uncertainties of phenological indicators of maize within the range of 5km to the station. The adopted approach performed well in extracting summer maize phenology. The correlations of v3 and maturity dates of summer maize are 0.60 and 0.80, respectively (Fig.7 c-d). Besides, the RMSE of v3 and maturity dates are 5.20 and 6.38 days. Nevertheless, the

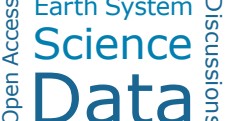

correlation of derived maturity dates of spring maize and corresponding records from AMS is relatively low, likely attributed to the discrepancies of the definitions between remotely sensed results and AMS. For instance, the v3 phase in AMS is defined as the date when the third leaf is exposed from the second leaf sheath, and the maturity is defined as the date when the dry weight of maize grains first reaches the maximum, more than 80% of the outer bracts of the plants turn yellow, the filaments

5   become dry, and the grains harden (Li et al., 2021). These definitions in AMS are challenging to be measured from remote sensing, and they are slightly different in terms of their definitions.





**Fig. 7:** Comparison of long-term mean phenology indicators derived from Landsat (satellites) and AMS (in-situ). The error bars of x- and y- axes represent uncertainty (i.e., one standard deviation) of multi-year phenological indicators and the mean phenological indicators within a certain extent (5km) of the AMS, respectively. (a-b) and (c-d) represent results from spring and summer maize, respectively.

Annual dynamics of derived phenology indicators (i.e., v3 and maturity) also agree well with the AMS observations (Fig.8). The comparison of annual results is similar to that from the long-term mean phenology. In general, the annual dynamics of phenological indicators in summer maize are better than that of spring maize, and this finding is consistent with previous studies (Huang et al., 2019a). The correlations of phenology indicators (i.e., v3 and maturity) of summer maize derived from Landsat and AMS are 0.34 and 0.59, respectively, notably higher than spring maize with corrections of 0.16 and 0.51. The difference between these two datasets is mainly attributed to (1) lacking accurate locations of the crop in the AMS data; (2) the crop planting patterns may be altered over the years (Fig. 9); and (3) variations from their definitions.

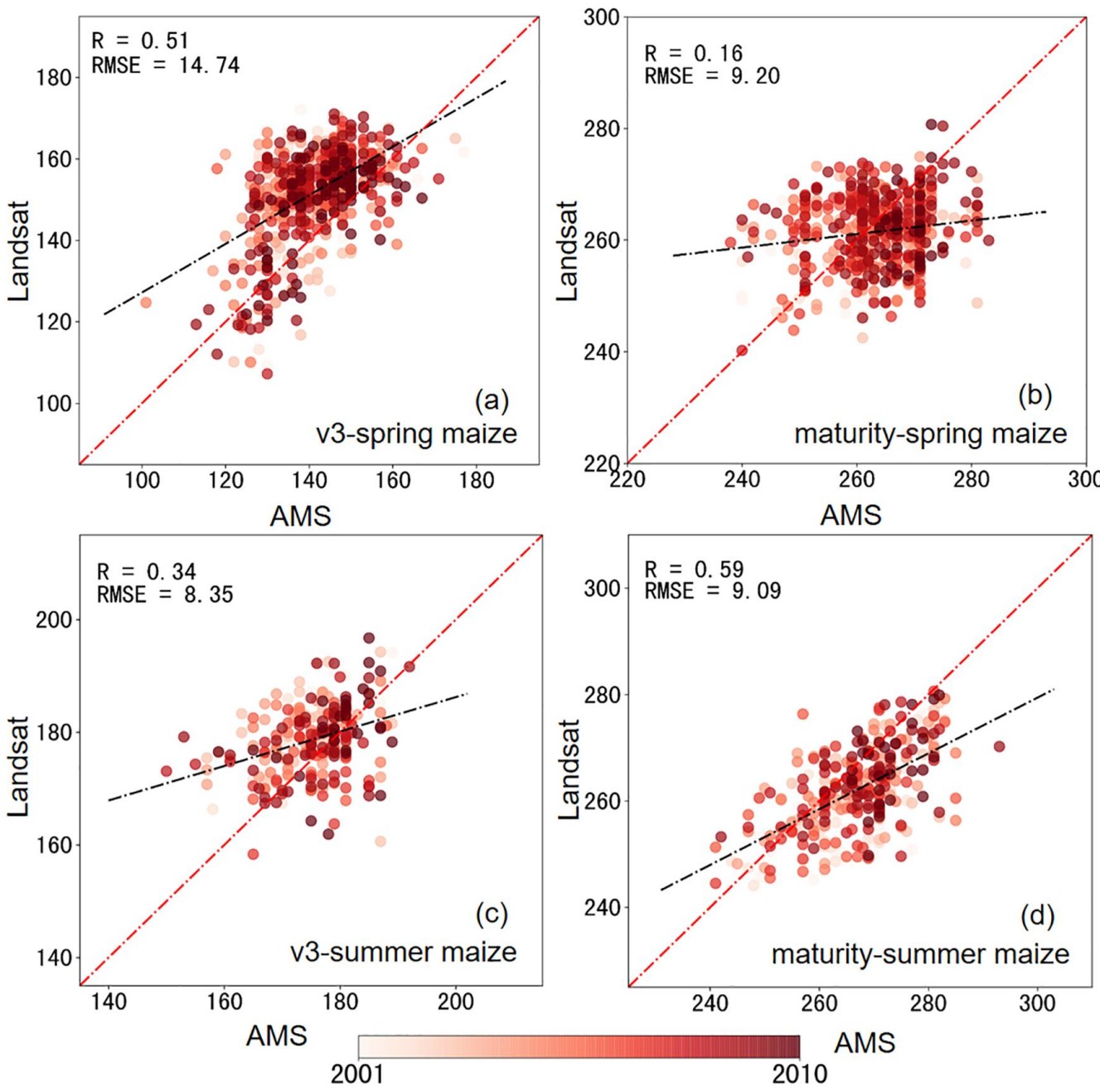

**Fig. 8:** Comparison of annual dynamics of derived phenology indicators from Landsat data and AMS observations from 2001 to 2010, including v3 and maturity of spring (a-b) and summer (c-d) maize.

**Fig. 9:** Cases with significant change of crop planting patterns. The blue ellipses indicate identified anomaly of EVI observations. From the ⓒ Google Earth images, we can note that the red dots are all located in the plots.

**4.3 Comparison with PhenoCam data**

5    Using the phenology mapping approach in this study, we observed a good agreement between Landsat derived and PhenoCam derived phenological indicators (Fig.10). We chose the United States (US) because no PhenoCam data are available in China. The same approach adopted in China for crop phenological indicator mapping was also used in the US with agriculture sites



where PhenoCam data are accessible. Thus, the feasibility of our approach can be evaluated. The phenology dataset provided by Richardson et al., (2018) is collected by continuous observations of vegetation growth by digital cameras. PhenoCam sites in Fig. 10 are mainly distributed in agriculture ecosystems, with records spanning from 2015 to 2018. Definitions of phenology indicators from Landsat and PhenoCam are consistent, i.e., definitions of *transition_10* and *transition_50* date when VI series

5   data crossed10% and 50% EVI2 amplitude (Richardson et al., 2018). The correlations of v3 and maturity dates from Landsat and PhenoCam are 0.74 and 0.63, respectively, with the root mean square error (RMSE) of 7.61 (v3) and 7.11 (maturity) days. Observations from these two datasets are located around the 1:1 line, suggesting the adopted mapping approach of phenology from satellite data can well match the in-situ observations. Possible reasons behind explaining their difference can be attributed to (1) different vegetation indices were used (i.e., EVI and GCC) and (2) the scale effect caused by their data sources.

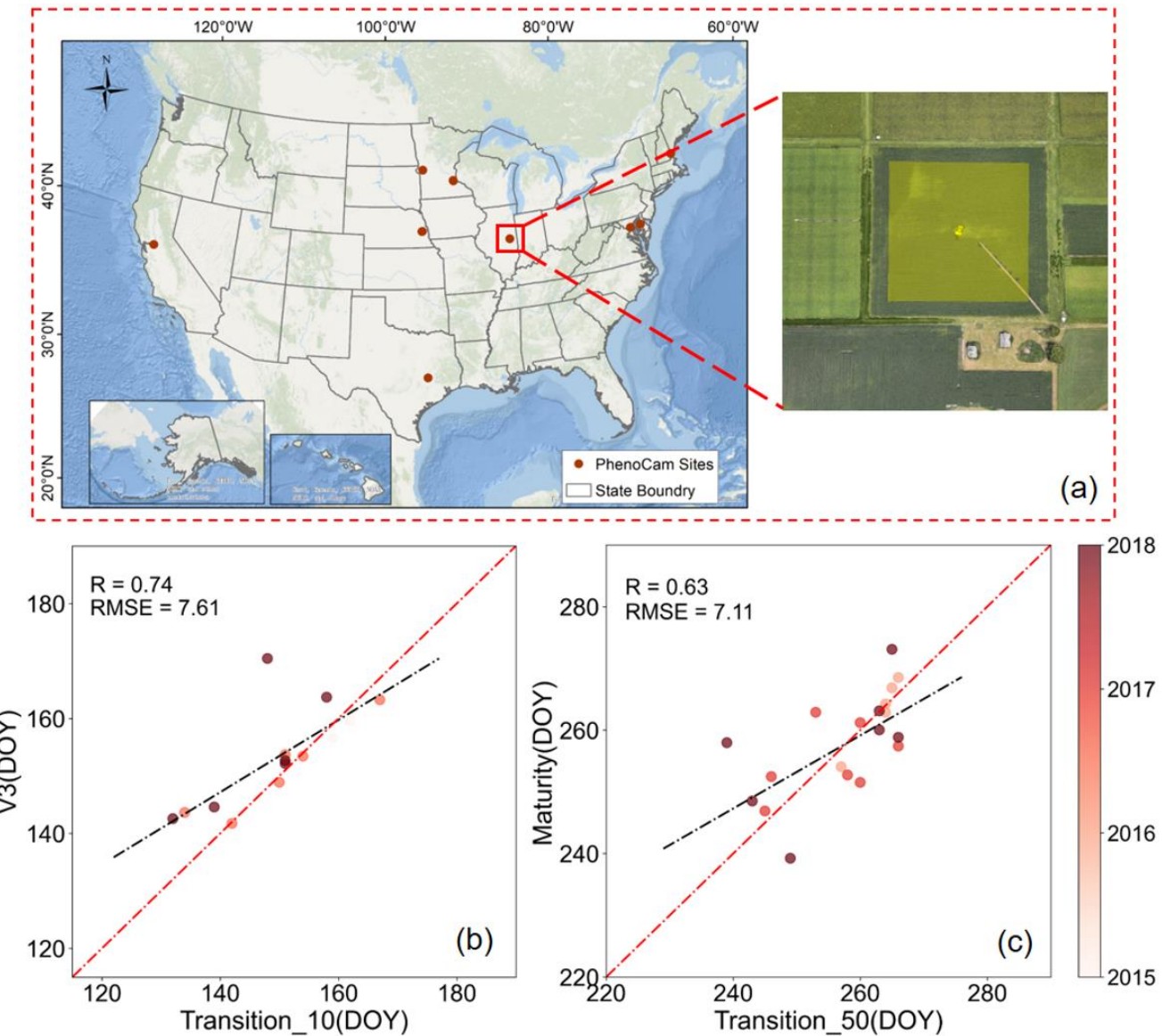

**Fig. 10:** Selected PhenoCam sites in agriculture ecosystems (a). The comparison of annual v3 (b) and maturity (c) dates were compared between the Landsat and PhenoCam derived results from 2015 to 2018. The basemap is provided by ESRI (https://www.arcgis.com/apps/mapviewer/index.html).



## 4.4 Comparison with MODIS phenology dataset

 The derived phenology indicators from Landsat and MODIS have a consistent temporal trend (Fig.11). MODIS phenology product (MCD12Q2) provides multiple phenology indicators (e.g., midgreendown) with two vegetation cycles. For summer maize with two vegetation cycles, we selected the phenology indicators of the second cycle for comparison. In the green-down

segment of one crop cycle, the MCD12Q2 product provides three phenology indicators, i.e., dormancy, midgreendown, and senescence, defined as 90%, 50%, and 10% of the segment EVI2 amplitude in a specific cycle. We selected the midgreendown indicator in the MODIS phenology product to compare in this study because it has the same definition as the maturity date in Landsat-derived results. We aggregated the fine-resolution maize data to the same resolution of MODIS and only kept those relatively pure pixels (maize pixels accounting for more than 50% of them) for comparison. We found the temporal trends of

derived phenology indicators of spring and summer maize from Landsat images are consistent with those derived from MODIS data in most years (Fig.11b). Our approach can well capture the dynamics (i.e., delay and advancement) of the crop growth phases, with magnitude difference between maturity date derived from Landsat observations and midgreendown derived from MCD12Q2 is within three days in most years. Different data sources and fitting methods likely cause discrepancies between the two phenology datasets. In addition, it is worthy to note that there is a high correlation of the maturity dates derived from

Landsat and MODIS (Fig.11c).

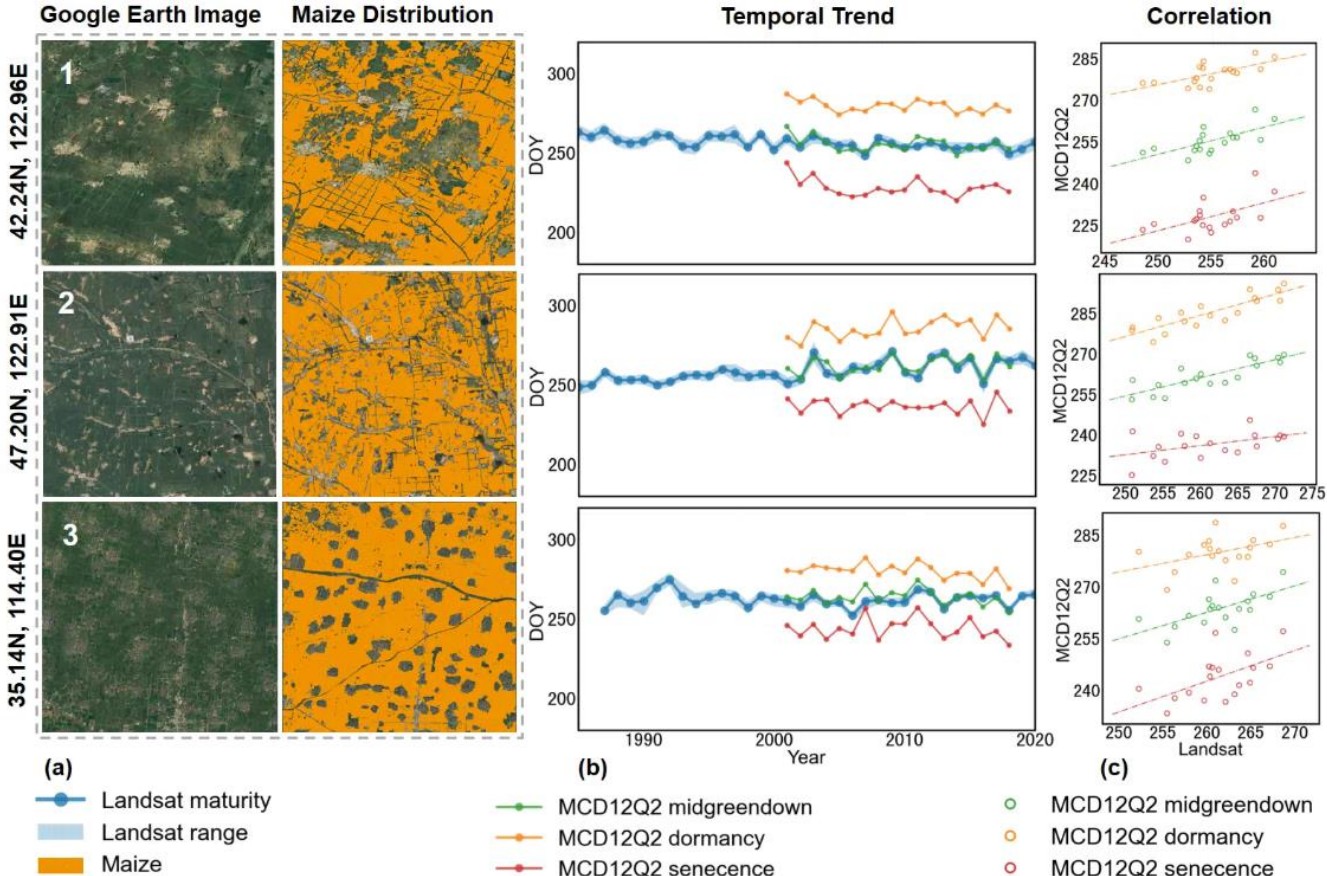

**Fig. 11:** Representative cases of phenology comparison between MODIS and Landsat derived results. Selected cases of maize (including the raw ⓒ Google Earth images and the distribution of maize) were displayed in (a), with comparison of their temporal trends (b) and corrections (c). Cases 1-2 are the spring maize and case 3 is the summer maize. Each scene represents a 1.5km x 1.5km square. Note that solid lines represent the mean phenology at the regional scale, and the shadowed areas represent the range from $25^{th}$ to $75^{th}$ quantiles of maturity date derived from Landsat.

Phenology indicators derived from Landsat observations also have a close spatial pattern to the MODIS phenology product but more spatial details (Fig.12). For example, there is a noticeable advancement of maturity in 2018 and a delay in 2015 (red boxes in Fig. 12), and these variations are captured by the two phenology datasets successfully. Besides, we can note that Landsat-derived phenology indicators (e.g., maturity) depict the difference of crop growth stages with more spatial details compared to the MODIS phenology product.

**Fig. 12:** Comparison of the maturity date in Landsat and the midgreendown date in MODIS from 2001 to 2018. The selected scene is the

case 1 in Fig 5. Red boxes are highlighted regions where these two products have noticeable difference.

### 4.5 Analyze with climate data

5    The summer maize is more sensitive to temperature and precipitation than the spring maize (Fig. 13). We used the monthly

mean air temperature and the total precipitation from April to October (Peng et al., 2019) in the Beijing-Tianjin-Hebei region

for analyses from 2011 to 2020. Overall, the mean total precipitation and mean temperature change range within the spring maize planting areas are larger than in the summer maize. Meanwhile, summer maize growing areas are mainly distributed in high-temperature areas (i.e., above 20℃). These results suggest the summer maize is more sensitive to surrounding climate conditions than spring maize.

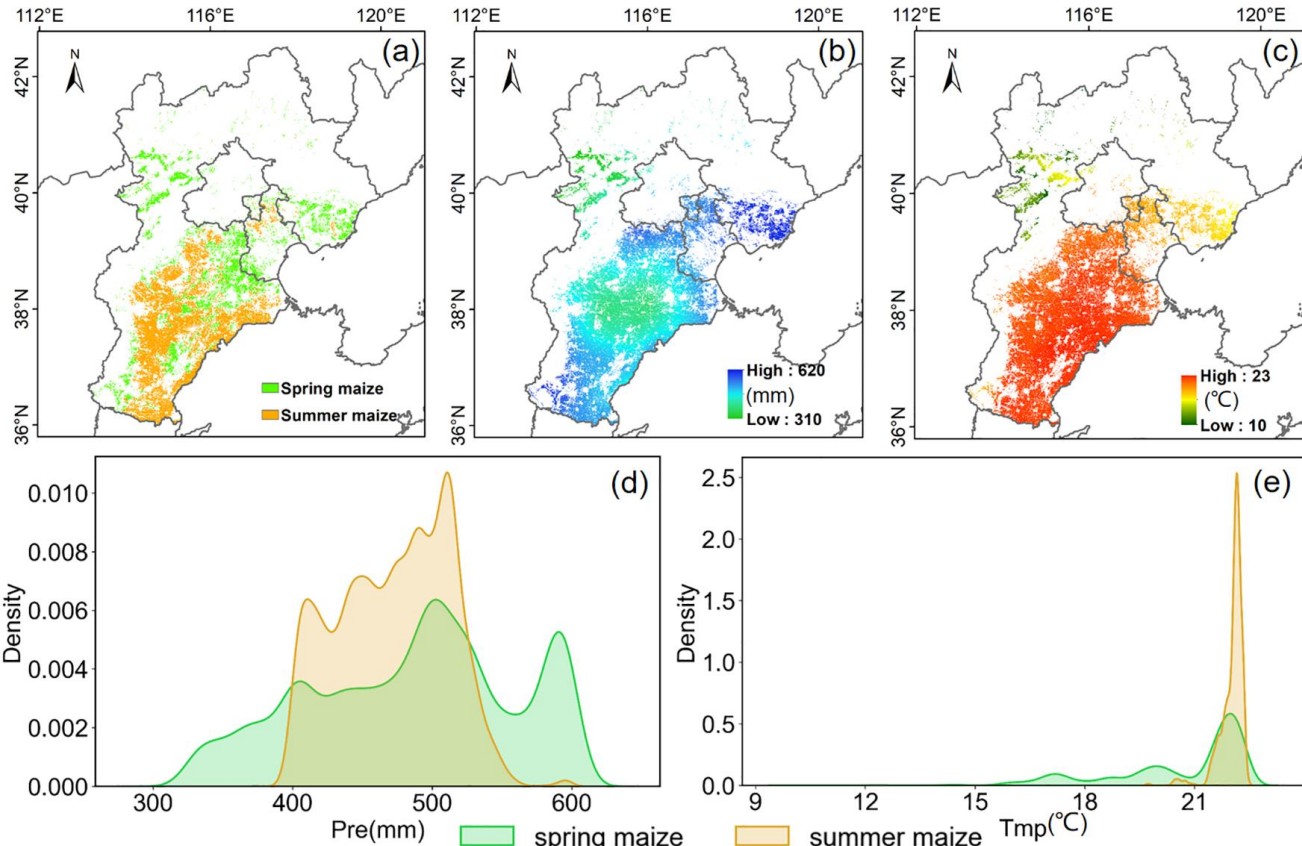

**Fig. 13:** The spatial distribution of spring and summer maize (a), mean total precipitation (b) and mean temperature (c) during the growing period of crop (from April to October) from 2011 to 2020. Kernel density curves of the mean annual total precipitation and temperature in the Beijing-Tianjin-Hebei region are provided in (d) and (e), respectively.

We observed a noticeable difference in the temporal trends of the derived maize phenology indicators before and after 2000

10 (Fig. 14). The temporal trends of derived phenology indicators, including v3 and maturity date of spring and summer maize before and after 2000, are notably different. The v3 shows an upward trend before 2000, whereas the maturity shows a



downward trend. However, the temporal trends of derived phenology indicators after 2000 are different, i.e., the v3 date is advanced, and the maturity date is delayed. For climate variables, the temperature within maize planting areas shows an upward tendency over the years. However, the mean total precipitation shows different trends before and after 2000, although the overall temporal trends are not explicit. The annual dynamics of maize phenology indicators may be partly attributed to the

5    rising temperature and annual variations of total precipitation. It was worth noting that the response of maize phenology and growth season duration was not comprehensively considered in this study.





**Fig. 14:** The temporal trends of phenological indicators (i.e., v3, maturity) and climate variables (i.e., mean temperature and mean total precipitation) during the growing period (from April to October), from 1985 to 2020. Two segments (i.e., 1985~2000 and 2001~2020) were independently fitted due to their distinct difference of temporal trends.

## 5 Data availability

This dataset provides annual dynamics of maize phenological indicators with a fine spatial resolution (30m) and a long temporal span (1985-2020) in China. The extracted phenology indicators include v3 (the date when the third leaf is fully expanded) and maturity (when the dry weight of maize grains first reaches the maximum). The format of this dataset is GeoTiff, with a spatial reference of WGS84. Each file in this dataset is named based on the information of phenology indicator, start year, end year, and the province. We divided the maize phenology of the entire time period into three parts: 1985-2000, 2001-

2010, 2011-2020 (Table 1). In total, we included 17 provinces in our study, i.e., Beijing, Tianjin, Hebei, Henan, Shanxi, Shaanxi, Shandong, Hubei, Anhui, Jiangsu, InnerMongolia, Ningxia, Gansu, Xinjiang, Heilongjiang, Jilin, and Liaoning. The derived    annual    maize    phenology    data    in    China    from    1985    to    2020    are    available    at https://doi.org/10.6084/m9.figshare.16437054 (Niu et al., 2021).

Table 1. Detailed band information in each formation.

| Band name | Year | Content | Range |
|-----------|------|---------|-------|
| Band1 | start year | | |
| Band2 | start year + 1 | maize phenology | 1-365 |
| … | … | | |
| Band N-1 | end year | | |
| Band N | | maize type | 1 - spring maize, 2 - summer maize |

Note: The range of phenology was set between 1 and 366 for leap years.

## 6 Conclusions

In this study, we generated the first annual maize phenology product with a fine spatial resolution (30m) and a long temporal span (1985-2020) in China, using all available Landsat images on the GEE platform. First, we extracted long-term mean phenology indicators (including v3 and maturity) from multi-year Landsat observations using the harmonic model. Second,

we identified the annual dynamics of phenological indicators by measuring the difference of dates when the EVI in specific years equals the fitted value.

The maize phenology product derived from Landsat scenes shows a good agreement with the commonly used phenology dataset. Our derived maize phenology datasets consistently meet the in-situ observations from the AMS and the PhenoCam phenology network. In addition, the phenology dataset in this study has similar temporal trends and can provide more spatial details compared to the MODIS phenology product. In addition, we observed a noticeable difference in the temporal trend of maize phenology before and after 2000, which is likely attributable to increasing temperature and annual variations of precipitation.

The extracted maize phenology dataset has great implications for crop field management and studies of the response of maize phenology to the changing environment. There are noticeable differences in crop growth due to diverse local climates, soil properties, and anthropogenic activities (such as sowing dates). The derived phenology product with a fine spatial resolution can delineate the difference and provide corresponding information to improve the field management and yield estimation (Zeng et al., 2020; Bolton and Friedl, 2013). In addition, this phenology product can also be used to investigate the response of crop phenology to global warming (Badeck et al., 2004; Niu et al., 2021). However, this study does not consider land cover changes (e.g., urban expansion and planting system change), which needs to be further investigated. For example, the maize distribution was regarded as persistent in our study over past decades.

**Author contributions**

Quandi Niu, Xuecao Li, and Jianxi Huang designed the research, performed the analysis, and wrote the paper; Hai Huang, Xianda Huang, Wei Su, and Wenping Yuan revised the manuscript.

**Competing interests**

The authors declare that they have no conflict of interest.





**Acknowledgments**

This study was supported by the National Natural Science Foundation of China (41971383).

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
