# Peer review of "A 30-m annual maize phenology dataset from 1985 to 2020 in China"

_Earth System Science Data, 2021_

## Author Response (AR1)

**Response to comments**

**Title:** A 30-m annual maize phenology dataset from 1985 to 2020 in China

**MS No.:** ESSD-2021-343

5    **Reviewer #1:**

**Comment #1-1.** This manuscript demonstrates the applied methods to produce a nationwide and long-term of dataset on maize phenology in China. The authors verified the accuracy of the Landsat derived product by comparing the estimated phenological parameters to those obtained from various types of data sources (i.e., ground truth, PhenoCam and MODIS). A dataset which covers large area and time span of phenological information is crucial for government officials and researchers to have a better

10   understanding of the nationwide dynamic changes in crop phenology and their drivers. However, I don't think the paper can be accepted by ESSD in its present form.

**Response:** Thank you for your comments that help to improve our manuscript. We carefully revised our manuscript thoroughly and provided a point-by-point response below.

15   **Comment #1-2.** Lack of novelty and interest. The authors follow the basic procedures/steps to generate the phenological dataset, which has already been widely available for different sites and countries around the world. I don't see any challenges for the methods applied and new findings from the analysis of the generated dataset. To improve its originality and make the paper more interesting, the authors could further analyze the spatial variabilities in maize phenology and to what extent they relate to weather conditions spatially, rather than only a simple comparison from the trend plots of phenology and weather

20   conditions as shown in Fig14.

**Response:** Thank you. Although various studies have been reported about approaches used to extract phenology metrics using satellite observations, most of them focus on moderate and coarse resolutions. Existing fine-resolution phenology studies were conducted in the urban domain. In contrast, limited attention has been paid to the agricultural fields, where it is common that the crops are associated with multiple cycles within a year. In this research, we generated the first national maize phenology

25   product with a fine spatial resolution (30m) and a long temporal span (1985-2020) in China, using the harmonic model. The harmonic model can delineate multiple cycles of crop rotations compared with the commonly used double logistic model (Li et al., 2019). We clarified these issues in our revised manuscript.

*"Compared with medium-resolution satellite data such as MODIS, the Landsat satellite data can provide numerous land surface records from 1985 to the present, which helps derive the long-term crop phenology dynamics. Unfortunately, limited*

 *attempts have been made using Landsat data to map the crop phenology with a fine resolution and a long-term span in China due to the complex planting patterns (Luo et al., 2020; Wu et al., 2010). Also, the computing resources required for such a mapping project are a huge challenge (Dong et al., 2016)." (Section 1, Paragraph 3)*

Further analysis of the spatial variabilities in maize phenology and the extent to which they relate to weather conditions is an interesting but challenging topic (especially the phenology of summer maize is influenced by the double-cropping system). We expanded the region of interest to the entire study area in this research and revised our manuscript. We observed a noticeable difference in the temporal trends of the climate variables and the derived phenology indicators before and after 2000 across the entire study area. We added our results in the revised manuscript as below:

*"We observed a noticeable difference in the temporal trends of the derived maize phenology indicators before and after 2000 (Fig. 14). The temporal trends of derived phenology indicators, including v3 and maturity date of spring and summer maize before and after 2000, are notably different. For climate variables, the temperature within maize planting areas has a steeper upward trend (the slope is more than 0.5℃ per decade) before 2000 than after 2000. The mean total precipitation shows different trends before and after 2000. It is worth noting that the precipitation within the spring maize producing area has a diverse and sharper tendency compared with that of the spring maize grown area. For phenological indicators, the changes in spring maize phenology mainly concentrated after 2000. The v3 date is advanced (-0.37 days per year), and the maturity date is delayed (0.38 days per year). The v3 and maturity indicators of summer maize have an advanced tendency before 2000, while the maturity date is delayed after 2000. The annual dynamics of maize phenology indicators may be partly attributed to the rising temperature and annual variations of total precipitation. In this research, we did not consider the impact of other factors (such as photoperiod and genotype of maize) on the variations of maize phenology and the response of maize phenology and growth season duration to climate change was not comprehensively considered." (Section 4.5, Paragraph 2)*

[Figure]

50

*Fig. 14: The temporal trends of phenological indicators (i.e., v3, maturity) and climate variables (i.e., mean temperature and mean total precipitation) during the growing period (from May to October), from 1985 to 2020. Two segments (i.e., 1985~2000 and 2001~2020) were independently fitted due to their distinct difference in temporal trends. We provided the temporal trend of variables across the study area and the interannual variations within different major agricultural zones.*

55 *Reference:*

- *Luo, Y., Zhang, Z., Chen, Y., Li, Z., and Tao, F.: ChinaCropPhen1km: a high-resolution crop phenological dataset for three staple crops in China during 2000–2015 based on leaf area index (LAI) products, Earth Syst. Sci. Data, 12, 197–214, https://doi.org/10.5194/essd-12-197-2020, 2020.*

- Wu, W., Yang, P., Tang, H., Zhou, Q., Chen, Z., and Shibasaki, R.: *Characterizing Spatial Patterns of Phenology in Cropland of China Based on Remotely Sensed Data*, Agric. Sci. China, 9, 101–112, https://doi.org/10.1016/S1671-2927(09)60073-0, 2010.

- Dong, J., Xiao, X., Menarguez, M. A., Zhang, G., Qin, Y., Thau, D., Biradar, C., and Moore, B.: *Mapping paddy rice planting area in northeastern Asia with Landsat 8 images, phenology-based algorithm and Google Earth Engine*, Remote Sens. Environ., 185, 142–154, https://doi.org/10.1016/j.rse.2016.02.016, 2016.

**Comment #1-3.** Unclear descriptions about the dataset (layers of polygons) used to identify maize farmland. The author should firstly clarify the basic attributes/forms of maize planting areas in the target research area China. This could be the common type of maize planting (smallholder farms or industrial agricultural system), average size of individual fields, or the management schemes, etc. This information is important to give readers a good overview of the maize planting system in the research area and the performance of the output product according to the given maize planting conditions. On top of this basic information, the authors should better explain the dataset (i.e., shapefile) used to identify maize farmland. The authors applied two data sources to delineate maize areas with different spatial resolution, which is not good for consistency and result in possible uncertainties subsequently. In addition, the dataset for defining the maize areas is not clearly stated, causing the following analysis less convincing. Alongside a better explanation of the dataset, perhaps the authors can also present the dataset in form of polygons on a map under a zoomed in view.

**Response:** Thank you for your comments that helped improve our manuscript. First, for clarity, we added descriptions about the polygons (i.e., maize farmland), transformed from classification results into vectors, and used them to indicate the cover of maize under a zoomed view in one specific site. We also provided the basic attributes of maize planting areas in our revised manuscript.

*"The spring maize is mainly distributed in Northeast China, dominated by the single cropping type. However, summer maize is mainly planted in the Huang-Huai-Hai Plain (Fig. 1c), where the double cropping system (rotation between winter wheat and summer maize) is commonly seen (Luo et al., 2020). In addition, there is also a certain amount of maize in other provinces (e.g., Xinjiang province). The growth period of summer maize spans roughly from June (after the harvest of winter wheat) to October compared with that of spring maize from May to October. Furthermore, the maize in Northeast China is mainly rain-fed. In contrast, the irrigation is needed for maize commonly exists in Huang-Huai-Hai Plain and Northwest China (arid and semi-arid areas) (Wu et al., 2010)" (Section 2, Paragraph 1).*

Second, both products mark the pixels with the largest proportion of maize. The national crop classification result is still scarce in China at present. The maize mapping product based on the approach of "temporal similarity assessment" in Dong et al. (2020) has a relatively larger extent than the results in You et al. (2021). Still, the latter product in Northeast China is retrieved using a machine learning method and has higher accuracy than the former. So the maize map in You et al. (2021) was adopted

to show the spatial distribution of maize in Northeast China. We applied two data sources to delineate maize areas, although this behavior may cause some errors. For clarity, we explained this issue in our revised manuscript as below.

*"You et al. (2021) derived the crop maps in Northeast China using a random forest, with optimized features including spectral, temporal, and textural characteristics (gray-level co-occurrence matrix). In other provinces, the maize maps were obtained using the "temporal similarity assessment" approach proposed by Dong et al. (2020). The distribution of maize is mainly determined by comparing the similarity of the vegetation index series of unknown pixels with a referred curve derived from maize fields. The retrieved maize datasets both have been validated with massive survey data with reliable performance (Fig. S1). The accuracy of the maize map in Northeast China is 0.85 (more than 8000 samples for cross-validation in 2019), and that of maize maps in other provinces is 0.79 (about 2000 samples for validation). Given that the original resolutions of these two classification maps are 10m (i.e., Northeast China) and 30m (i.e., other provinces), we aggregated the 10m maize map to 30m in our study." (Section 2, Paragraph 3).*

[Figure]

*Fig. 1: Spatial distribution of maize within the study area (a), which contains 17 provincial-level administrative regions (b). The green polygons transformed from classification result in pixel form in (a) indicate maize's cover in one specific site. Subplot c shows the nine*

105 *agricultural zones in China, and the data is from the Institute of Geographic Sciences and Natural Resources Research, Chinese Academy of Sciences. In addition, the base map of figures is from ESRI (https://www.arcgis.com/apps/mapviewer/index.html)*

Third, we added a zoomed view to show the detailed classification results in our revised supplement.

[Figure]

**Fig. S1:** *Distribution of maize with in the study area and we provided the detailed maize map at six different sites. The base map of the figure is from ESRI.*

*Reference:*

- *Luo, Y., Zhang, Z., Chen, Y., Li, Z., and Tao, F.: ChinaCropPhen1km: a high-resolution crop phenological dataset for three staple crops in China during 2000–2015 based on leaf area index (LAI) products, Earth Syst. Sci. Data, 12, 197–214, https://doi.org/10.5194/essd-12-197-2020, 2020.*

- *Wu, W., Yang, P., Tang, H., Zhou, Q., Chen, Z., and Shibasaki, R.: Characterizing Spatial Patterns of Phenology in Cropland of China Based on Remotely Sensed Data, Agric. Sci. China, 9, 101–112, https://doi.org/10.1016/S1671-2927(09)60073-0, 2010.*

- *You, N., Dong, J., Huang, J., Du, G., Zhang, G., He, Y., Yang, T., Di, Y., and Xiao, X.: The 10-m crop type maps in Northeast China during 2017–2019, Sci. Data, 8, 41, https://doi.org/10.1038/s41597-021-00827-9, 2021.*

**Comment #1-4.** The manuscript should be better polished. I find a lot of typos, and descriptions that are hard to understand. It's easy to distract readers from the content itself. I would suggest the authors to get editing help from someone with full professional proficiency in English.

**Response:** Thank you. As suggested, the language in this version has been polished by native speakers with trackable edits.

**Reviewer #2:**

**Comment #2-1.** This article aims to provide a long-term national maize phenology dataset with a high spatial resolution. The adopted method and newly released dataset should have good application value for crop phenology monitoring and agricultural production management at different regional scales.

130 **Response:** Thank you for your positive comments. We carefully revised our manuscript and provided a point-by-point response below.

**Comment #2-2.** The maize distribution map may be more consistent compared with land use change map, but it is better to try harder to reduce the impact of the assumption that the maize distribution was regarded as persistent over 30 years, for 135 example, maybe using GEE to get maize classification maps.

**Response:** Thank you for your comments. Yes, we agree that our assumption that the maize distribution has been persistent over the past three decades is not true. The maize price and climate conditions change across different years, resulting in specific dynamic maize distribution. However, it is also worth noting that mapping the maize dynamics at the national scale is a challenging task, which relies on the collection of massive samples for training and validation and numerous satellite 140 observations. For instance, the available Landsat images are lower in earlier years than in recent years, during which period the Sentinel-2 data can also be used. There is no public available maize dynamic product in China with fine spatial resolution and a long temporal span. Given that the dynamics of maize distribution are beyond our scope, we focus on the maize phenology dynamics rather than their extents in this study. For clarity, we explained this issue in our revised manuscript as below.

145 *"It is worth note the maize distribution map is consistent across different years in our study due to the relatively stable planting situation as one of the major crops(Sun et al., 2007; Li et al., 2008). Of course, we also admitted that certain dynamics exist in maize distribution due to the changing maize price, climate conditions, and choice of farmers across different years. Mapping the maize dynamics at the national scale in China is a challenging task because of the scarcity of massive ground samples. There is also no public available maize dynamic product with fine spatial resolution and a long temporal span. So 150 we kept the maize distribution map consistent and derived dynamics of maize phenology indicators with tolerable errors."* *(Section 2, Paragraph 3)*

*Reference:*

- *Sun, H., Zhang, X., Chen, S., Pei, D., and Liu, C.: Effects of harvest and sowing time on the performance of the rotation of winter wheat–summer maize in the North China Plain, Ind. Crops Prod., 25, 239–247, 155 https://doi.org/10.1016/j.indcrop.2006.12.003, 2007.*

● *Li, H., Zheng, L., Lei, Y., Li, C., Liu, Z., and Zhang, S.: Estimation of water consumption and crop water productivity of winter wheat in North China Plain using remote sensing technology, Agric. Water Manag., 95, 1271–1278, https://doi.org/10.1016/j.agwat.2008.05.003, 2008.*

160 **Comment #2-3.** From Fig.8, it is not as described that the correlations of two phenology indicators of summer maize is significantly higher than that of spring maize, or the author may put the wrong figure here.

**Response:** Thank you for your comments. We are sorry for this confusion. In general, the derived summer maize phenology data has a higher accuracy when compared with spring maize phenology (especially at maturity phases). And we have rephrased our description in our revised manuscript.

165 *"Annual dynamics of derived phenology indicators (i.e., v3 and maturity) also agree well with the AMS observations (Fig.8). The comparison of annual results is similar to that of the long-term mean phenology. In general, the annual dynamics of phenological indicators in summer maize are better than that of spring maize (especially at maturity phases), and this finding is consistent with previous studies (Huang et al., 2019a). The correlations of phenology indicators (i.e., v3 and maturity) of summer maize derived from Landsat and AMS are 0.34 and 0.59, respectively, notably higher than spring maize with*
170 *corrections of 0.16 and 0.51. And for spring maize, the correlations of v3 and maturity indicators from the two datasets are 0.51 and 0.16."* *(Section 4.2, Paragraph 2)*

[Figure]

*Fig. 8: Comparison of annual dynamics of derived phenology indicators from Landsat data and AMS observations from 2001 to 2010, including v3 and maturity of spring (a-b) and summer (c-d) maize.*

175 **Reference:**

- *Huang, Liu, Zhu, Atzberger, and Liu: The Optimal Threshold and Vegetation Index Time Series for Retrieving Crop Phenology Based on a Modified Dynamic Threshold Method, Remote Sens., 11, 2725, https://doi.org/10.3390/rs11232725, 2019a.*

**Comment #2-4.** Fig.13 does not show that summer maize is more sensitive to temperature and precipitation than spring maize, and this description needs to be supported by quantitative evidence or scientific findings.

**Response:** Thank you. As suggested by the reviewer, we expanded the analysis in the Jing-Jin-Ji region to the entire national scale. As such, our descriptions of the analyses have been significantly updated. We provided the distribution proportion of different types of maize in several agricultural zones and the corresponding meteorological conditions of two kinds of maize planting areas. Details can be found in our revised manuscript below.

*"Summer maize has a higher requirement for hydrothermal conditions (especially for temperature) than spring maize (Fig. 13). Northeast China Plain and Huang-Huai Hai Plain (Fig. 1c) are the two largest maize-producing areas in China, about 60% of spring maize is grown in the Northeast China Plain (Fig. 13b), and more than 80% of summer maize is distributed in Huang-Huai-Hai Plain (Fig. 13c). In addition, we used the monthly mean air temperature and the total precipitation from May to October (Peng et al., 2019) in the study area for analyses from 2011 to 2020. Overall, the mean total precipitation and mean temperature change range within the spring maize planting areas are larger than in the summer maize. Meanwhile, summer maize growing areas are mainly distributed in high-temperature areas (i.e., above 20℃). These results suggest that summer maize has a higher requirement for hydrothermal conditions than spring maize." (Section 4.5, Paragraph 1)*

[Figure]

195  *Fig. 13: The spatial distribution of maize across the study area (a) and pixel values represent the coverage of maize in one kilometer. Figures*
*(b) and (c) indicate the coverage of spring and summer maize. Additionally, we also provided the proportion of maize in major agricultural*
*zones. The mean total precipitation (b) and mean temperature (c) during the growing period of the crop (from May to October) from 2011*
*to 2020 are presented in (d) and (e). And figures (f) and (g) show the kernel density curves of the mean annual total precipitation and*
*temperature in the study area. The abbreviations are as follows: NCP: Northeast China Plain; LP: Loess Plateau; Nor: Northern arid and*
200  *semiarid region; HHH: Huang-Huai-Hai Plain; MLYP: middle-lower Yangtze Plain.*

*Reference:*

● *Shouzhang Peng, Ding, Y., Liu, W., and Li, Z.: 1 km monthly temperature and precipitation dataset for China from 1901*
*to 2017, Earth Syst. Sci. Data, 11, 1931–1946, https://doi.org/10.5194/essd-11-1931-2019, 2019.*

205  **Comment #2-5.** Considering that there are about 10 provinces where summer maize is grown, it may be required to expand
the region of interest instead of only selecting Beijing-Tianjin-Hebei area to reveal the impact of climate materials for deeper
analysis and better persuasion.

**Response:** Thank you for your comments that helped improve our manuscript. As suggested, we expanded the region of
interest to the entire study area and revised our manuscript. The spatial distribution of spring and summer maize is analyzed
210  in our response to Comment #2-4, supported by spatial variabilities of climate factors such as temperature and precipitation.
Besides, we also analyzed the temporal trends of these factors over the past decades. We observed a noticeable difference in

the temporal trends of the climate variables and the derived phenology indicators before and after 2000 across the entire study area. Details can be found in our revised manuscript below.

*"We observed a noticeable difference in the temporal trends of the derived maize phenology indicators before and after 2000 (Fig. 14). The temporal trends of derived phenology indicators, including v3 and maturity date of spring and summer maize before and after 2000, are notably different. For climate variables, the temperature within maize planting areas has a steeper upward trend (the slope is more than 0.5 ℃ per decade) before 2000 than after 2000. The mean total precipitation shows different trends before and after 2000. It is worth noting that the precipitation within the spring maize producing area has a diverse and sharper tendency compared with that of the spring maize grown area. For phenological indicators, the changes in spring maize phenology mainly concentrated in the segments after 2000. The v3 date is advanced (-0.37 days per year), and the maturity date is delayed (0.38 days per year). The v3 and maturity indicators of summer maize have an advanced tendency before 2000, while the maturity date is delayed after 2000. The annual dynamics of maize phenology indicators may be partly attributed to the rising temperature and annual variations of total precipitation. In this research, we did not consider the impact of other factors (such as photoperiod and genotype of maize) on the variations of maize phenology and the response of maize phenology and growth season duration to climate change was not comprehensively considered."* (Section 4.5, Paragraph 2)

[Figure]

Fig. 14: The temporal trends of phenological indicators (i.e., v3, maturity) and climate variables (i.e., mean temperature and mean total precipitation) during the growing period (from May to October), from 1985 to 2020. Two segments (i.e., 1985~2000 and 2001~2020) were independently fitted due to their distinct difference in temporal trends. We provided the temporal trend of variables across the study area and the interannual variations within different major agricultural zones.

---

## Referee Report (RR1)

Dear Editor,

I have read the revised version of manuscript "A 30-m annual maize phenology dataset from 1985 to 2020 in China". The authors have done a great effort to improve the manuscript and addressed most of my comments. With some minor edits, I agree to publish this paper in ESSD. Please find my detailed comments in the Comments to Authors.

Best regards

**Comments to Authors**
The authors have made good responses to my concerns, while strengthening the paper. I have re-read the manuscript and have some detailed comments. In my view the paper merits publication in ESSD after a minor revision.

Detailed comments:
- Fig14: indicate the two phenological indicators: v3 and maturity in both the figure and caption.
- Fig1: try to simplify the figure and caption. I don't see rich information for readers from Fig1(b), and (c). Try to integrate them into one or two subfigures.
-- FigS1: with in -> within